# The Genomic landscape of short tandem repeats across multiple ancestries

Prashanth Vijayaraghavan[1], Sergey Batalov[1], Yan Ding[1], Erica Sanford[1,2], Stephen F. Kingsmore[1], David Dimmock[1], Charlotte Hobbs[1], Matthew Bainbridge[1]*

1 Rady Children's Institute for Genomic Medicine, San Diego, CA, United States of America, 2 Cedars-Sinai Medical Center, Los Angeles, CA, United States of America

☯ These authors contributed equally to this work.
* MBainbridge@rchsd.org

**Data Availability Statement:** Data are within a FigShare repository, which can be accessed via the following URL: https://figshare.com/articles/dataset/All_STR_Positions/21759434.

## Abstract

Short Tandem Repeats (STRs) have been found to play a role in a myriad of complex traits and genetic diseases. We examined the variability in the lengths of over 850,000 STR loci in 996 children with suspected genetic disorders and 1,178 parents across six separate ancestral groups: Africans, Europeans, East Asians, Admixed Americans, Non-admixed Americans, and Pacific Islanders. For each STR locus we compared allele length between and within each ancestry group. In relation to Europeans, admixed Americans had the most similar STR lengths with only 623 positions either significantly expanded or contracted, while the divergence was highest in Africans, with 4,933 chromosomal positions contracted or expanded. We also examined probands to identify STR expansions at known pathogenic loci. The genes *TCF4*, *AR*, and *DMPK* showed significant expansions with lengths 250% greater than their various average allele lengths in 49, 162, and 11 individuals respectively. All 49 individuals containing an expansion in *TCF4* and six individuals containing an expansion in *DMPK* presented with allele lengths longer than the known pathogenic length for these genes. Next, we identified individuals with significant expansions in highly conserved loci across all ancestries. Eighty loci in conserved regions met criteria for divergence. Two of these individuals were found to have exonic STR expansions: one in *ZBTB4* and the other in *SLC9A7*, which is associated with X-linked mental retardation. Finally, we used parent-child trios to detect and analyze *de novo* mutations. In total, we observed 3,219 *de novo* expansions, where proband allele lengths are greater than twice the longest parental allele length. This work helps lay the foundation for understanding STR lengths genome-wide across ancestries and may help identify new disease genes and novel mechanisms of pathogenicity in known disease genes.

## Introduction

A repetitive, tandem DNA pattern sequence involving a 1–6 base pair arrangement is known as a Short Tandem Repeat or STR [1]. STRs can be useful genetic tools due to their highly mutable nature and association with several known rare diseases and can be inherited in a

**Funding:** This was funded in part by a grant from NIH (R01HL145175) to Matthew Bainbridge.

**Competing interests:** The authors have declared that no competing interests exist.

dominant or recessive manner [2]. The polymorphisms in STRs are due to variability in the number of copies of the repeated arrangement in different individuals and populations [3]. STRs have long been used in the fields of forensic science and DNA profiling as a method of determining relatedness between individuals. This is due to both the polymorphic nature of STRs as well as their heritability from parents to offspring [4]. Most STR expansions are less than 100 base pairs, but they can also be much larger in length, spanning several hundred or thousand base pairs [5].

In this study, we examined four aspects of variability in the lengths of STRs. First, we examined the variability STR length between ancestral groups. There is a known association between STR polymorphisms and ancestral populations [6]. In previous studies, a small number of autosomal STRs have been used for the purposes of predicting an individual's ethnicity with varying degrees of certainty. However, these studies are limited to approximately a dozen autosomal STR sites, with very little data available regarding STR length variability beyond these loci [7]. Our focus in this part of the study was to examine the variability in STR lengths across six ancestral groups for nearly 850,000 different loci. Five of the six ancestries studied (Africans, East Asians, Europeans, Admixed Americans, and Non-admixed Americans) were grouped and organized according to ancestral informative markers (AIMs), while the sixth ancestry group, Pacific Islanders, was organized based on self-reported ancestry. The majority of Pacific Islanders showed no strong correlation to an existing ethnic group.

Next, STR expansions in known pathogenic loci were examined. While most STRs are in intragenic and/or noncoding regions of DNA, there are 50 different coding loci at which STR expansions have been causally associated with human diseases [8, 9]. Examples include *DMPK* (OMIM#605377) (CTG Expansion), within which a trinucleotide expansion of 50 repeat units (150 base pairs) can result in myotonic dystrophy (OMIM#160900), *TCF4* (OMIM#602272) (CTG Expansion), a gene associated with Fuchs endothelial corneal dystrophy (OMIM#613267) which is caused by an STR expansion of more than 150 base pairs, and *AR* (OMIM#313700) (GCA Expansion), in which a 114-base pair expansion within the gene results in muscular atrophy of Kennedy (OMIM#313200). In this part of the study, we examined probands with large expansions in 28 loci with known pathogeneses related to STR expansion.

For the third part of our study, we wanted to focus on STRs that occur in regions of minimal STR length variability between ancestral groups. These highly conserved regions have traditionally been understudied but their closer inspection may reveal new pathways for disease.

Finally, we examined *de novo* STR mutations (expansions and contractions), with the goal of expanding understanding of the role of *de novo* mutations in STRs, including their potential for previously unknown mechanisms of disease inheritance.

## Results

### Demographics

Our total data set of 2,193 individuals was organized into three partially overlapping study groups. A total of 1,946 individuals were in the Population cohort group. Demographic breakdowns of the Population cohort group can be found in **Table 1**. The Population cohort group was used to examine the variability in the length of STRs across different ancestral groups. This cohort also was used to calculate statistics (e.g., average, standard deviation, etc.) that would serve as baseline values for subsequent study groups. The population cohort was grouped by primary ancestry. Our second study group consisted only of Probands. This group was used to study both STR expansions in known pathogenic STR loci as well as individuals with highly divergent STR lengths in conserved regions of DNA. A total of 996 probands were

**Table 1. Demographic breakdown.**

| Group | Number of Individuals | Proband | Mother | Father |
|---|---|---|---|---|
| Population Cohort | 1946 | 768 | 616 | 562 |
| African | 161 | 82 | 40 | 39 |
| European | 1297 | 489 | 412 | 396 |
| East Asian | 104 | 41 | 38 | 25 |
| Nonadmixed Americans | 203 | 83 | 63 | 57 |
| Admixed Americans | 162 | 67 | 56 | 39 |
| Pacific Islanders | 19 | 6 | 7 | 6 |

included this group. Demographic breakdowns of this group can be found in **Table 2**. The final study group consisted only of Trios (families in which mother, father, and proband were all sequenced). A total of 184 families (552 individuals in total) were included in the Trio Cohort. Trios were used to identify *de novo* expansions and contractions in probands. Tables showing the complete ethnic breakdown for all individuals in our studies can be seen in **Tables A-C in S1 Table**.

## Variability in the length of STRs across ancestral groups

For each of the 850,000 chromosomal loci, the average European allele length was compared to average allele length in the five non-European populations to identify divergent positions. Absolute Z-score was the primary criteria used to identify divergent positions due to its ability to describe the relationship of a value to mean of a group of values. An absolute Z-score of 5 was chosen to isolate those positions which are extremely divergent from the mean. Lower Z-scores were considered but resulted in too many positions being labelled as divergent. To help normalize the data and mitigate short positions that may read as highly divergent, an additional criterion of a greater than 20% difference between a position's European average length and the position's comparative non-European average length was incorporated. Admixed Americans presented with the fewest number of divergent positions (623), followed by Pacific Islanders (1,560) and Non-admixed Americans (2,259). East Asians and Africans showed the largest number of divergent loci compared to Europeans, with 3,606 and 4,933 divergent positions respectively. Scatter plots displaying the divergent positions can be viewed in **Fig 1A–1E**.

The 20 most divergent loci in each non-European group (Africans, East Asians, Pacific Islanders, Admixed Americans, non-Admixed Americans) were isolated for further analysis. All of these positions were intronic and the vast majority were in genes with no known disease association. Of the 100 total positions examined, 14 were shared by at least two ancestries and one position, corresponding to the gene *RFC3*, was found in three ancestral groups (Pacific Islanders, Admixed Americans, and Non-admixed Americans). Thus, most of an ancestry's

**Table 2. Ethnic breakdown of probands.**

| Group | Number of Individuals |
|---|---|
| All Groups | 996 |
| Africans | 107 |
| Europeans | 643 |
| East Asians | 55 |
| Nonadmixed-Americans | 104 |
| Admixed Americans | 81 |
| Pacific Islanders | 6 |

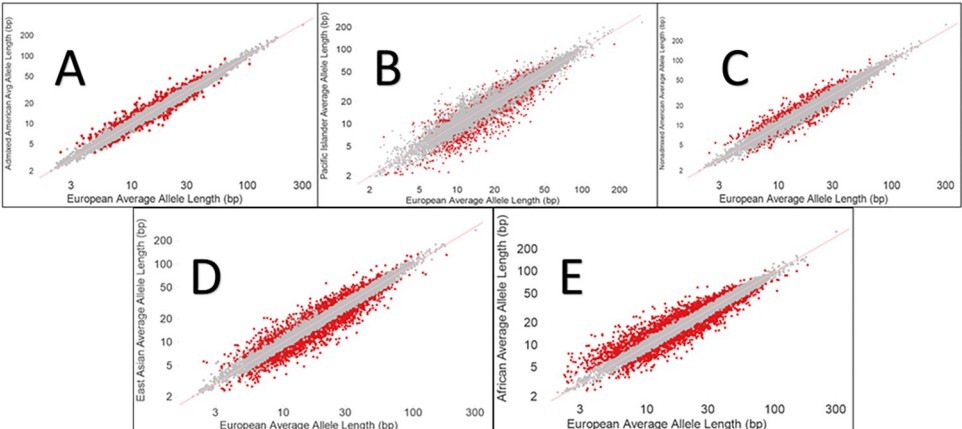

**Fig 1.** A-E. STR Length Variability Between Groups: Scatterplots displaying the ancestral variability in STR lengths when compared with Europeans. Each dot represents an STR locus. Red dots indicate STR loci whose lengths were determined to be divergent (Absolute Z-score > 5 and > 20% difference in length when compared with Europeans). Grey dots correspond to STR loci whose lengths did not meet the divergent threshold criteria. A: EUR vs Admixed AMR B: EUR vs PAC C: EUR vs Nonadmixed AMR D: EUR vs EAS E: EUR vs AFR.

divergent loci were unique to that ancestry. Positions with no associated genes are marked "N/A". These divergent loci are displayed in **Tables A-E in S1 Table**.

## STRs in known pathogenic loci

Twenty-eight STR-associated pathogenic loci were examined in all individuals in the Proband Cohort. These positions are loci where STR expansions are known to be associated with pathogenic manifestations, with each loci having a studied length of pathogenicity. Individuals with divergent positions were designated based on a 150% difference between an individual's allele length in one of these known pathogenic loci and the average total population allele length for that same loci. This cutoff was used instead of simply isolating those positions with expansions longer than the associated pathogenic length to have a balance of both likely pathogenic expansions as well as pre-mutation expansions to better understand the associated diseases. Positions in the genes *AR*, *TCF4*, *DIP2B* and *DMPK* showed the greatest number of divergent individuals with 162, 49, 23, and 11 people respectively meeting the divergence criteria. Amongst these individuals, allele lengths were longer than the cutoff for pathogenicity in 49 of 49 divergent *TCF4* (CTG Expansion) probands, 6 of 11 divergent *DMPK* (CTG Expansion) probands, none (0 of 23) *DIP2B* (GGC Expansion) probands, and none (0 of 162) divergent *AR* (GCA Expansion) probands. The six individuals with *DMPK*-associated allele lengths longer than the known pathogenic cutoff were known to have been molecularly and clinically diagnosed with myotonic dystrophy (OMIM#160900), an autosomal dominant disorder. Variants in *TCF4* are associated with Fuchs Endothelial Corneal Dystrophy (OMIM#613267,) a common autosomal dominant degenerative disorder with an incidence of nearly 5% of people over 40 years old living in the United States [10]. Fuchs Endothelial Corneal Dystrophy is incompletely penetrant, with Xu et al. indicating a penetrance of 86.2% in individuals between 64 and 91 years in age and a penetrance of 44.4% in younger individuals aged 19–46 years of age. The average age of diagnosis is 66 years old [9]. The probands in our study were all children, thus, as expected no probands with divergent positions in *TCF4* displayed the associated phenotype. Variants in *AR* have been found to cause spinal and bulbar muscular atrophy of Kennedy (OMIM#313200), an X-linked recessive form of spinal muscular atrophy, but none of the

**Table 3. STR expansions in pathogenic loci within proband cohort.**

| Gene | Number of Divergent Loci | Ethnic Breakdown (AMR:EUR:Other) | Pathogenic Length Reference (BP) | Number of Expansions Longer than Pathogenic Length |
|---|---|---|---|---|
| Total | 255 | 38:172:45 | N/A | 55 |
| AR | 162 | 32:108:22 | 114 | 0 |
| TCF4 | 49 | 3:38:8 | 150 | 49 |
| DMPK | 11 | 1:8:3 | 150 | 6 |
| DIP2B | 23 | 0:15:7 | 600 | 0 |
| ATXN10 | 2 | 2:0:0 | 4000 | 0 |
| RFC1 | 8 | 0:3:5 | 4000 | 0 |

This table displays genes corresponding to known pathogenic STR loci which appeared the most often in our proband cohort. Number of Divergent Loci indicates the number of times a divergent STR length (>150% compared to average length) appears in the proband cohort. The Ethnic Breakdown shows the number of loci corresponding to the American, European, and Other ancestral groups. The Pathogenic Length Reference is the known cutoff for pathogenic STR length in base pairs for a particular locus [11] The final column indicates how many loci are longer than the Pathogenic reference length.

divergent allele lengths in the probands were longer than the described pathogenic length. Violin plots displaying the distribution of allele lengths for each ancestral group in the proband cohort for *DMPK* and *TCF4* can be seen in **Figs A and B in S1 Fig**.

Longer allele lengths in two other genes with established relationship to disease, *RFC1* (OMIM#102579) (AAAG Expansion), associated with autosomal recessive cerebellar ataxia, neuropathy, and vestibular areflexia syndrome (OMIM#614575,) and *ATXN10* (OMIM#611150) (ATTCT Expansion), associated with autosomal dominant spinocerebellar ataxia 10 (OMIM#603516), were also present in the proband cohort. For *RFC1*, 8 individuals had allele lengths longer than the total population average allele length. An allele length between 155 and 175 base pairs was observed in all eight of these individuals. Two individuals had a divergent allele length for *ATXN10*, with both individuals' allele lengths at least 195 base pairs. All expansions in *RFC1* were heterozygous, which explains why there was no phenotypic match between the patients and the recessive disease associated with that gene. There was also no phenotypic crossover between those individuals who had expansions in *ATXN10* and its linked disease, however this is likely because the manifestations of autosomal dominant spinocerebellar ataxia 10 are adult onset. All genes where at least one proband displayed a divergent locus can be seen in **Table 3**.

Ancestral average allele length was also calculated for each of the 28 pathogenic related STR loci. We found no relationship between ancestry and average allele length or pathogenic distribution. This information is available **in S2 Table**.

## Divergent STR lengths in conserved regions of DNA

STR expansions in highly conserved regions of DNA are understudied and generally little is known about them. To find those positions which were the most conserved, a covariance < = 0.075 was used as a cutoff. Initially larger covariances > 0.15 were considered, but too many positions were then declared conserved. By using such a low covariance value, only the most conserved positions were isolated. After identifying the conserved regions, individual positions with divergent lengths were defined as those greater than 3 standard deviations from the mean, resulting in a total of 80 divergent positions. This strict cutoff for divergence helped to provide those expansions which were most anomalous. Of those eighty positions, 43 positions also had a percent difference from the total average of greater than 250%, meaning these positions were highly divergent from the mostly conserved regions. Forty-one of the positions were intronic and had no known associated disease.

**Table 4. Individuals with divergent STR lengths in conserved regions of DNA.**

| Chromosome | HG19 Position | Individual | Reference Length (bp) | Longest Allele Length (bp) | Total Population Average Length (bp) | Covariance | Gene | Region |
|---|---|---|---|---|---|---|---|---|
| 18 | 10742797 | 02CAM | 30 | 106 | 31.460 | 0.060 | *PIEZO2* | Intron |
| 15 | 64029600 | 9AMYP | 16 | 52 | 15.950 | 0.070 | *HERC1* | Intron |
| 21 | 44281469 | 43C2K | 16 | 52 | 16.010 | 0.040 | *WDR4* | Intron |
| 17 | 7369806 | F889J | 12 | 36 | 11.995 | 0.036 | *ZBTB4* | Exon |
| X | 46618390 | 32KVE | 12 | 33 | 12.003 | 0.035 | *SLC9A7* | Exon |

Two positions occurred in the exons of genes of known disease association, although in neither case is STR expansion a known mechanism of disease. One position occurred within exon three (of four) in *ZBTB4* (OMIM#612308) (TGTC Expansion), an ethyl-CpG-dependent transcription repressor, but there is no known relationship to human disease at this time. The other exonic position occurred within exon one (of seventeen) in *SLC9A7* (OMIM#300368) (TC Expansion), which is associated with Intellectual developmental disorder, X-linked 108 (OMIM#301024). This X-linked recessive disease is characterized by significant intellectual disability, developmental delay, and variable physical manifestations including dysmorphic facies, pes planus, hypotonia, and fifth finger clinodactyly. Our patient received Rapid Whole Genome Sequencing as a neonate and because the most significant phenotypic characterizations of this disorder are the intellectual disability and developmental delay and since the associated disease is recessive, we are not sure if there is phenotypic overlap. Both of these individuals had their expansions validated in the clinical laboratory via Sanger Sequencing.

*PIEZO2* (OMIM#613629) (TA Expansion), *HERC1* (OMIM#605109) (AATA Expansion), and *WDR4* (OMIM#605924) (CA Expansion) are conserved, disease related genes unassociated with STR expansion. A single individual with one divergent intronic position was found in our cohort for each of these three. *PIEZO2* is associated with autosomal dominant and autosomal recessive distal arthrogryposis (OMIM#114300,108145,617146), *HERC1* variants have been linked to autosomal recessive Macrocephaly, dysmorphic facies, and psychomotor retardation (OMIM#617011), and *WDR4* is associated with autosomal recessive Galloway-Mowat syndrome 6 (OMIM#618347) as well as an autosomal recessive syndrome of microcephaly, growth deficiency, seizures, and brain malformations (OMIM#618346). However, none of these disease descriptions were found to phenotypically align with the probands' clinical presentations. Divergent positions can be observed in **Table 4**.

### *De novo* STR mutations

Proband STR loci with allele lengths greater than two times the lengths of each parent's longest allele were flagged as potential *de novo* expansion sites, while proband allele lengths with half the length of either parent's shortest allele were flagged as potential *de novo* contraction sites. The average number of *de novo* expansions per proband was 17, with a range of 1 to 40. Neither paternal or maternal age nor ethnicity were associated with the number of *de novo* of expansions. (**Figs A-C in S2 Fig**) In total, we observed potential 3,219 *de novo* expansions and 9 potential *de novo* contractions. All 9 contractions and 3,001 of 3,219 expansions occurred in the introns of genes without known relationship to disease.

There were 26 *de novo* expansions in the exons of non-disease associated genes. Interestingly, of these 26 expansions, the genes *APPL2* (TG Expansion) and *HEG1* (GCA Expansion) were frequent sites of expansion, with 14 and 5 instances of *de novo* expansions respectively. In many of these instances, the proband allele length was at least 300% longer than that of the parent.

## Validation

A subset of 20 expansions were selected for validation. Fourteen positions were believed to be potential *de novo* expansions and six positions were expansions that were believed to occur in conserved regions of DNA. Of the 14 potential *de novo* expansions, only four expansions were found to be true *de novo* expansions. Of the other ten positions determined not to be *de novo*, one was discovered to be a deletion when validated. Four expansions were present in both parents and five expansions were present in one parent, indicating that these positions were also not true *de novo* expansions. All fourteen positions were validated via Long Reads. Twelve of these *de novo* positions were also validated via Sanger Sequencing and two positions were validated by gel electrophoresis.

All six expansions that occurred in conserved regions of DNA were validated as true expansions. All six of these expansions were validated via Long Reads and Sanger Sequencing. All validations can be seen **in the S3 Table.**

# Discussion

Here we assessed STRs in a genome-wide fashion in a large cohort with diverse ancestral backgrounds and a range of pediatric disease presentations.

## Variability in length of STRs across ancestral groups

Our data demonstrate a correlation between STR length and ancestral population for several STR loci. As anticipated, Admixed Americans presented with the fewest number of divergent loci when compared with Europeans, while Non-admixed Americans had more divergent loci, which is consistent with the fact that the Admixed American population shares a great number of genetic markers with the European population. The African population had the largest number of divergent loci, which is consistent with the well-reported considerable genetic diversity in African versus other ancestral populations [12].

Historically, disproportionate representations of Europeans in STR studies have a left an incomplete picture regarding the true breadth of polymorphisms in STRs, especially in understudied ethnic groups. Studying these STRs with an ancestral context may help to discover polymorphisms within and amongst ethnic groups, potentially even being used as a tool for a population's predisposition to certain STR-related pathogeneses or as a grouping tool for ethnicities in population genetics.

## STRs in known pathogenic loci

Of the 28 genes with known STR-associated pathogeneses examined in our cohort, *TCF4* and *DMPK* were the only two genes for which probands were found to have allele lengths longer than the known pathogenic length. A total of 49 individuals (5% of the proband cohort) were found to have allele lengths longer than the *TCF4* pathogenic length of 150 base pairs. The associated disease, Fuchs Endothelial Corneal Dystrophy (OMIM#613267), is a late-onset, autosomal dominant disorder with a 5% incidence in individuals over 40 years old which is consistent with our findings ($P = 0.65$, Pearson's chi-X Test) [10]. Despite the allele lengths, given our patients' ages we do not know if the probands will manifest the FECD phenotype. *DMPK*-associated myotonic dystrophy (OMIM#160900) is among one of the best-studied STR loci. In our proband cohort, 6 individuals were found to have at least one allele longer than 150 base pairs, the pathogenic length for *DMPK*. All 6 patients had been diagnosed with myotonic dystrophy. Based on our results, identifying allele length divergence in known pathogenic STR loci can aid in molecular diagnostic yield.

### Divergent STR lengths in conserved regions of DNA

STR expansions in conserved regions of DNA are not only rare but are also incredibly under-studied. We identified 35 positions that were highly conserved across all ancestries and yet were considerably expanded in at least one subject. Twenty-seven of these positions (77%) occurred in the intronic regions of genes with no known disease association. We also found an STR expansion in the first exon of *SLC9A7* in one proband and another STR expansion in exon three of *ZBTB4* in another proband. Although none of these findings were diagnostic for those patients it is likely novel STR expansion loci will be identified through increased use of WGS as a diagnostic tool.

### *De Novo* STR mutations

Short Tandem Repeats are a common etiology of *de novo* disease-causing mutations in people [13]. By better characterizing these *de novo* STRs, new pathogenic associations and disease mechanisms can be discovered. It has previously been reported that probands with Autism Spectrum Disorder exhibited a higher number of *de novo* STR mutations, with these expansions being longer and more evolutionarily deleterious [14]. ASD-associated STR mutations were seen more frequently with older paternal age. We did not observe any correlation between ancestry or parental age and the number or severity of STR mutations. By using the methods outlined in the study, *de novo* STR expansions and contractions can be more easily and readily identified. Focusing on these *de novo* mutation sites may help identify new disease genes and novel mechanisms of pathogenicity in known disease genes.

### Limitations

Our analysis had several limitations. The STR genotyping application used, GangSTR, is fairly robust in its ability to call out STR loci using short reads, however, it is limited by read and DNA fragment length [5, 14]. Thus, GangSTR cannot reliably call STR related pathogenic loci for which the known pathogenic length is greater than 150 base pairs in our WGS data.

The ancestry informative marker (AIM) panel used in this study is also a potential limitation. The AIM panel examines 250 genetic markers from 16 subpopulations and although it is more robust than panels used in similar studies, it still leaves out several subpopulations and even some superpopulations. The lack of informative markers for Pacific Islanders in this panel was the rationale behind using self-reported ancestry to identify the PAC ancestral group in this study, despite the inherent limitations of self-reported ancestry.

## Materials and methods

### Complete data set

This study was approved by the University of California San Diego IRB (#160468) and the WCG IRB (#20171726). All subjects in this study had written parental consent approving the use of their data and their child's data for research purposes.Our total data set consisted of 2,193 individuals from 1,523 families. The entire cohort contained 996 probands, 1,178 parents (616 mothers), and 19 siblings. 184 trios (mother, father, and proband of the same family) are another subset of our total dataset. WGS from the manufactured DBS were aligned to human genome assembly GRCh37 (hg19) and variants identified with the Illumina DRAGEN (Dynamic Read Analysis for GENomics) Bio-IT Platform (Illumina, v.3.4.5 and 3.7.5). Samples were run using different versions of DRAGEN with all samples having > = 30x coverage and probands all in the range of 35-50x. A verification process was applied to ensure the quality of DRAGEN variant calling as it pertains to Rady Children's Institute for Genomic Medicine

(RCIGM) clinical diagnostic standards. All probands were originally sequenced due to clinical indication. Probands were primarily patients from the neonatal Intensive Care Unit (ICU), but also included the pediatric ICU, hospitalized non-ICU patients, and outpatients with metabolic, neurological, and other suspected genetic disorders. This data was sequenced in house by Rady Children's Institute for Genomic Medicine.

### Data processing

All BAM files were processed through GangSTRv2.4 resulting in an output VCF for each BAM. GangSTR, a genome wide genotyping STR application which can identify over 850,000 STR loci in an individual, was the primary tool used for analyzing STR lengths [2]. Repeat expansion lengths called by GangSTR can vary, but the majority of expansions called are less than 6 base pairs. GangSTR works by extracting information from paired end reads into a uniform model to predict maximum likelihood tandem repeat lengths [2]. These gangSTR filters make use of enclosing read pairs, spanning read pairs, flanking read pairs, and fully repetitive read pairs to estimate STR lengths.

Although GangSTR is one of the most robust STR tools available, it is limited by read and DNA fragment length for detecting STR expansions over ~150bp [5, 14]. Next, call level filters were applied to the GangSTR output VCF files using dumpSTR (parameters:—filter-span-bound-only,—filter-badCI,—min-call-DP 50,—max-call-DP 1000,—min-call-Q 0.90). Using these filters, calls with abnormally high coverage, calls with no supporting reads, calls where only spanning or bounding reads are called, and calls for which the maximum likelihood genotype falls outside of the 95% bootstrap confidence interval were removed [15]. All GangSTR and DumpSTR scripts used in this study can be found in **S1 Document.**

Information regarding genes, diseases/disorders, and DNA region were annotated to the DumpSTR filtered VCFs using Cassandra (https://www.hgsc.bcm.edu/software/cassandra). Phen2Gene [16] was used to rank the candidate genes based on human phenotype ontology (HPO) terms. The entire workflow can be observed in **Fig 2**.

### Ancestry

In this study, we extended a previously published genetic admixture classifier, created by Wang et al., that uses an ancestry informative marker (AIM) set of 250 biallelic human SNPs within exonic regions (AIM250). The choice of the marker positions in exonic regions allows genetic admixture classification from both WGS and WES clinical samples. The past classifier was modified to add the fourth superpopulation reference representative genomes and to allow inference of the 4-way genetic admixture from four distinct and selective continental populations (African (AFR), European (EUR), East Asian (EAS) and Americans (AMR)). For this, the publicly available dataset from the 1000 Genomes Project database was iteratively classified using only AIM250 genomic positions with the goal of removing admixed or misclassified individual genomes and determining the core ancestral least admixed or indigenous inhabitants of the Panamerican supercontinent. Not unexpectedly, this training procedure evicted all members of Puerto-Rican (PUR) population as being strongly admixed with Caucasian cohort (EUR), while Peruvian (PEL) and a part of Mexican (MXL) populations formed a compact cluster (N = 148). For evenness of superpopulation, larger superpopulations were downsampled to the max of N = 300 representatives. The American cohort was further divided into two groups: Non-admixed Americans (> 75% AMR markers) and Admixed Americans, resulting in five ancestries based on AIM panels. If an individual had > 60% of the genetic markers for a certain ancestry, that ancestry would be assigned as that individual's primary ancestry. Individuals with between 50% - 60% of the genetic markers for a certain ethnicity would not only have a primary

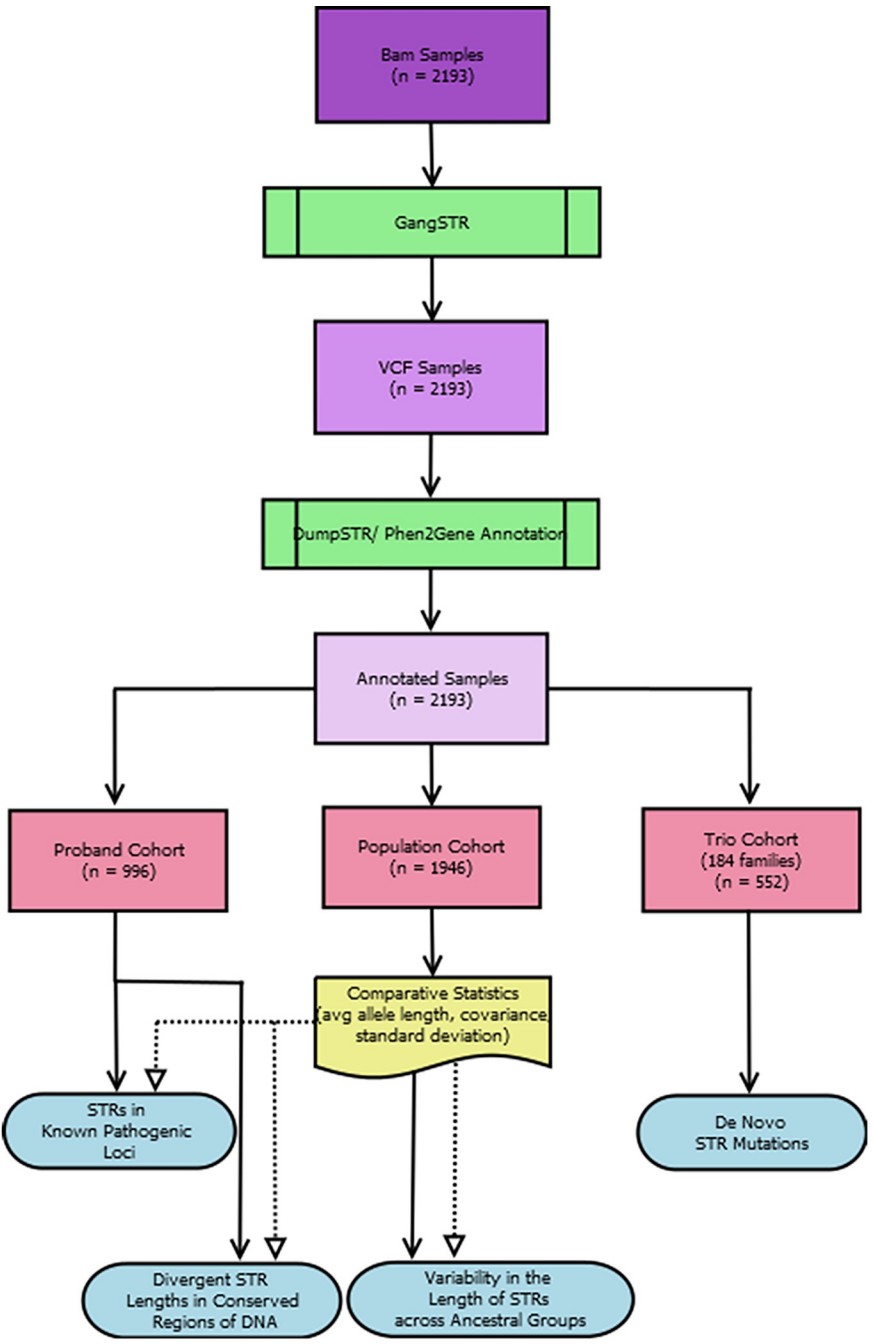

**Fig 2. Dataset workflow: Bam samples were processed through GangSTR, filtered by DumpSTR, annotated by Casandra/GeneRanks, then broken out into the Population cohort, proband cohort, and Trio Cohort.**

ancestry, but were also assigned a secondary ancestry, designated by the ancestry with the minority of genetic markers. A final group, Pacific Islanders (PAC), was also included and was assigned based on self-reported ethnicity. Nearly all of the Pacific Islanders presented with a combination of European and East Asian biomarkers from our AIM panel.

The genetic admixture classifier software is available from a public repository [https://github.com/radygenomics/AIM250-FOUR-CLADE]. Starting from a typical WGS or WES

variant file (VCF), the inference of the admixture from the four superpopulations is inferred and reported as a 4-vector with a linear norm of 1.

## Study populations

Our total data set of 2,193 individuals was organized into three partially-overlapping study groups: Population cohort, Proband cohort, and Trio cohort. The Population Cohort included all parents and only those probands whose parents were not in our complete data set. The Proband cohort consisted exclusively of probands. Our final study group consisted only of Trios (families in which mother, father, and proband were sequenced).

## Variability in length of STRs across ancestral groups

For the 1,946 individuals in the Population Cohort, allele tally, mean, and population standard deviation was calculated for each of the six ancestries (AFR, EAS, EUR, PAC, Admixed AMR, Non-admixed AMR) at each of the nearly 850,000 STR loci.

Based on the demographic breakdown of this group, the vast majority of individuals were of primarily European decent. Because of this all other ancestries were compared to those whose primary ancestry was European. For every position, comparative statistics, such as ethnic/ total average length, ethnic/ total standard deviation, Z-test comparative scores, and standard error, were calculated.

Z-score, which provides a quantitative relationship between a single value and a mean of a group of values, was used as the primary metric in determining divergent positions. In order to isolate those positions believed to be the most divergent, an absolute Z-score of 5, meaning a position's ethnic average must be at least +/- 5 standard deviations away from the position's European average, was determined as the cutoff. To help normalize the data and mitigate short positions that may read as highly divergent, an additional criterion of a > 20% difference between a position's European average length and the position's comparative non-European average length was incorporated.

## STRs in known pathogenic loci

Average allele lengths derived from our population cohort were used as a comparative metric in this part of our study. Twenty-eight different known STR-related pathogenic loci were then isolated in our proband cohort. Proband allele length was compared to the average allele length for these pathogenic loci. Proband loci which had greater than 250% difference compared to the mean population allele length were considered to be divergent.

## Divergent STR lengths in conserved regions of DNA

Average allele length, covariance, and standard deviation were calculated for each locus in our Population Cohort. Covariance, a measure in the relationship between two random variables, was used as a metric to determine whether or not a position was conserved. Low covariance values have been used to identify conserved protein residue interactions, and a similar approach was used in this study to classify conserved regions of DNA. Loci with a covariance < = 0.075 were considered conserved.

Once the conserved regions were identified, a methodology to isolate divergent lengths within these loci was needed. Since it was determined that loci were already conserved, the threshold for "divergent" did not have to be as stringent as in the comparison of ancestral groups. Instead, a proband was considered divergent at a conserved locus if their allele length was >3 standard deviations and at least 250% larger than the population mean.

### *De Novo* STR mutations

The trio cohort, which consisted of 184 families (proband, mother, and father) was used to find and analyze potential *de novo* STR expansions and contractions. For expansions, at each position the longest alleles from the mother and father were compared to the longest proband allele length. Proband allele lengths greater than two times the longest parental allele were isolated as potential *de novo* STR expansions. Positions at which the proband's longest allele length was determined to be less than half of the shortest allele length compared to both maternal and paternal alleles were determined to be potential *de novo* STR contractions.

## Conclusion

Studying the variability in STR lengths in ancestral populations can be an effective tool in examining both polymorphisms between and within groups and also potentially provide an avenue for a better understanding the molecular diagnosis, penetrance, and ancestral epidemiology of several genetic pathogeneses, STR-related or otherwise. Further studies are needed to aid in better diagnoses, greater knowledge of ancestral predisposition of certain diseases, and identification of new potential mechanisms of action for specific genetic disorders.

Some of the clinical data used in this study are not publicly available due to the Health Insurance Portability and Accountability Act, however the data that supports the findings of this study is available in a DNA Nexus Repository. In order to access this repository please create a DNA Nexus account and visit the following url: [https://platform.dnanexus.com/panx/projects/G6vJyPj0BYFv2KBJ4f55jkgf/data/]

## Supporting information

**S1 Fig. Allele length distribution for each ancestral proband group for *DMPK* and TCF4.** A: *DMPK* B: *TCF4.*
(TIF)

**S2 Fig. Correlation between parental age and number of expansions observed.** A. Correlation between Mother's age and number of expansions. B. Correlation between Father's age and number of expansions.
(TIF)

**S1 Table. STR Length Variability Between Ancestries: A: EUR vs Admixed AMR B: EUR vs PAC C: EUR vs Nonadmixed AMR D: EUR vs EAS E: EUR vs AFR.**
(DOCX)

**S2 Table. Pathogenic STR length by ancestry this table shows the most common pathogenic STR loci in the proband cohort.** * This position corresponds to two different genes.
(DOCX)

**S3 Table. Validations.**
(XLSX)

**S1 Document. GangSTR and DumpSTR scripts.**
(DOCX)

## Author Contributions

**Conceptualization:** Prashanth Vijayaraghavan, Sergey Batalov, Matthew Bainbridge.

**Data curation:** Prashanth Vijayaraghavan, Sergey Batalov, Matthew Bainbridge.

**Formal analysis:** Prashanth Vijayaraghavan, Matthew Bainbridge.

**Funding acquisition:** Matthew Bainbridge.

**Investigation:** Prashanth Vijayaraghavan, Matthew Bainbridge.

**Methodology:** Prashanth Vijayaraghavan, Matthew Bainbridge.

**Project administration:** Prashanth Vijayaraghavan, Matthew Bainbridge.

**Resources:** Prashanth Vijayaraghavan, Sergey Batalov, Matthew Bainbridge.

**Software:** Prashanth Vijayaraghavan, Sergey Batalov.

**Supervision:** Charlotte Hobbs, Matthew Bainbridge.

**Validation:** Yan Ding.

**Visualization:** Prashanth Vijayaraghavan.

**Writing – original draft:** Prashanth Vijayaraghavan.

**Writing – review & editing:** Prashanth Vijayaraghavan, Erica Sanford, Stephen F. Kingsmore, David Dimmock, Charlotte Hobbs.

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
