## [Decision Letter · Decision Letter 0]

30 Aug 2022

PONE-D-22-22440The Genomic landscape of short tandem repeats across multiple ancestriesPLOS ONE

Dear Dr. Vijayaraghavan,

Thank you for submitting your manuscript to PLOS ONE. After careful consideration, we feel that it has merit but does not fully meet PLOS ONE’s publication criteria as it currently stands. Therefore, we invite you to submit a revised version of the manuscript that addresses the points raised during the review process.

We look forward to receiving your revised manuscript.

Kind regards,

Alvaro Galli

Academic Editor

PLOS ONE

"This was funded in part by a grant from NIH (R01HL145175) to MNB."

 "This was funded in part by a grant from NIH (R01HL145175) to Matthew Bainbridge."

Please include your amended statements within your cover letter; we will change the online submission form on your behalf."

Additional Editor Comments (if provided):

Reviewers' comments:

Reviewer's Responses to Questions

**Comments to the Author**

1. Is the manuscript technically sound, and do the data support the conclusions?

Reviewer #1: No

Reviewer #2: Yes

2. Has the statistical analysis been performed appropriately and rigorously? 

Reviewer #1: No

Reviewer #2: No

3. Have the authors made all data underlying the findings in their manuscript fully available?

Reviewer #1: No

Reviewer #2: No

4. Is the manuscript presented in an intelligible fashion and written in standard English?

Reviewer #1: Yes

Reviewer #2: Yes

5. Review Comments to the Author

Reviewer #1: This paper performed whole genome sequencing on 2193 individuals to reveal the genomic landscape of short tandem repeats. The work has great potential significance. The authors can still further improve this paper.

1. The analysis in the manuscript is too simple comparing to the huge data they used. In the first part, they can try to find some AIM-STRs from the 850,000 STRs on the 6 groups. They can also perform some population genetic analysis based on the STR data (e.g. Principal component analysis). In the second part, the authors examined probands to identify STR expansions at known pathogenic loci. However, the results presented is not very clear. I suggest authors to summarize the results to a figure/table or both.

2. Line 151. Which figure is quoted here is not clearly described.

3. It is unclear the raw data is obtained from publicly available database or sequenced in the authors’ lab.

4. The data availability to the pubic should be provided.

5. The tables provided is not in a standard format and the data is not summarized as a scientific paper.

Reviewer #2: My concern is about M&M section... where authors just have written the names of software but didnt provide any script how these will work with this type of data... I would suggest to give scripts as ESM which will not only make MS more sound but attract more citations

6. PLOS authors have the option to publish the peer review history of their article (what does this mean?). If published, this will include your full peer review and any attached files.

Reviewer #1: No

Reviewer #2: No

---

## [Author Response · Author response to Decision Letter 0]

28 Sep 2022

Journal Requirements

• Please ensure that your manuscript meets PLOS ONE's style requirements, including those for file naming

o Our manuscript was edited to fit with PLOS ONE’s style and file naming requirements

• Please include your full ethics statement in the ‘Methods’ section of your manuscript file.

o Our ethics statement can now be found in the ‘Method’ section of our manuscript file.

• Please remove any funding-related text from the manuscript and let us know how you would like to update your Funding Statement

o An amended funding statement can now be found in the cover letter for this revised submission

• We note that you have stated that you will provide repository information for your data at acceptance. Should your manuscript be accepted for publication, we will hold it until you provide the relevant accession numbers or DOIs necessary to access your data.

o The information repository containing all data used in this study can be found in DNA Nexus and can be accessed via the following url: https://platform.dnanexus.com/projects/G6vJyPj0BYFv2KBJ4f55jkgf/data/

• Please include captions for your Supporting Information files at the end of your manuscript, and update any in-text citations to match accordingly.

o Supporting information captions are now included at the end of the manuscript

Review Comments

• The analysis in the manuscript is too simple comparing to the huge data they used. In the first part, they can try to find some AIM-STRs from the 850,000 STRs on the 6 groups. They can also perform some population genetic analysis based on the STR data (e.g. Principal component analysis).

o Although we agree it is an intriguing idea to conduct PCA on the STR data and we considered using some machine learning models to see if we could differentiate ethnicities based off of STR only we ultimately decided that that work is beyond the scope of this publication. 

• In the second part, the authors examined probands to identify STR expansions at known pathogenic loci. However, the results presented is not very clear. I suggest authors to summarize the results to a figure/table or both.

o We agree that there was some ambiguity in the results section dedicated to presenting the data on STR expansions in known pathogenic loci of probands. In response, that section of the manuscript has been modified to give more clarity. Additionally, Table 3, which summarizes the data contained in this section has been changed in order to better present the data gathered. 

• Line 151. Which figure is quoted here is not clearly described.

o The supplemental figure which is referenced in this line has been properly labelled.

• It is unclear the raw data is obtained from publicly available database or sequenced in the authors’ lab.

o An additional statement is now included in the ‘Methods’ section which clarifies that the raw data was sequenced by Rady Children’s Institute for Genomic Medicine

• The data availability to the pubic should be provided.

o The information repository containing all data used in this study can be found in DNA Nexus and can be accessed via the following url: https://platform.dnanexus.com/projects/G6vJyPj0BYFv2KBJ4f55jkgf/data/

• The tables provided is not in a standard format and the data is not summarized as a scientific paper.

o Changes have been made throughout the paper to better present our findings. Tables have also been modified to fit the standard format.

• My concern is about M&M section... where authors just have written the names of software but didnt provide any script how these will work with this type of data... I would suggest to give scripts as ESM which will not only make MS more sound but attract more citations.

o We acknowledge that it would be greatly beneficial if we provided the scripts used for GangSTR and the DumpSTR filters. To address this, we have included a supplemental document containing all pertinent scripts.

---

## [Decision Letter · Decision Letter 1]

4 Nov 2022

PONE-D-22-22440R1The Genomic landscape of short tandem repeats across multiple ancestriesPLOS ONE

Dear Dr. Vijayaraghavan,

Thank you for submitting your manuscript to PLOS ONE. After careful consideration, we feel that it has merit but does not fully meet PLOS ONE’s publication criteria as it currently stands. Therefore, we invite you to submit a revised version of the manuscript that addresses the points raised during the review process.

We look forward to receiving your revised manuscript.

Kind regards,

Alvaro Galli

Academic Editor

PLOS ONE

Reviewers' comments:

Reviewer's Responses to Questions

**Comments to the Author**

1. If the authors have adequately addressed your comments raised in a previous round of review and you feel that this manuscript is now acceptable for publication, you may indicate that here to bypass the “Comments to the Author” section, enter your conflict of interest statement in the “Confidential to Editor” section, and submit your "Accept" recommendation.

Reviewer #1: (No Response)

Reviewer #3: (No Response)

2. Is the manuscript technically sound, and do the data support the conclusions?

Reviewer #1: Yes

Reviewer #3: Partly

3. Has the statistical analysis been performed appropriately and rigorously? 

Reviewer #1: Yes

Reviewer #3: I Don't Know

4. Have the authors made all data underlying the findings in their manuscript fully available?

Reviewer #1: Yes

Reviewer #3: Yes

5. Is the manuscript presented in an intelligible fashion and written in standard English?

Reviewer #1: Yes

Reviewer #3: Yes

6. Review Comments to the Author

Reviewer #1: The STR loci were generated by GangSTR, however, the data validation is not performed as required from genomic sequencing data. At least the STR data should be confirmed by a different approach/tool for analyzing STR lengths and the selected samples should sequenced by Sanger sequencing for the further validation.

Reviewer #3: Thank you for asking me to review this manuscript. This indicates that expansion propensity varies depending on different haplotypes. I think that a study of ethnic diversity or of how a haplotype may correlate with a tendency to expansion is interesting. However we would require more detailed/systematic clinical data to see for example if these expansions may also explain some of the morbidity. But although the numbers of the cohorts mentioned in the manuscript are impressive, I am afraid that my expertise is not appropriate for making a judgment about the soundness of the Methods used.

7. PLOS authors have the option to publish the peer review history of their article (what does this mean?). If published, this will include your full peer review and any attached files.

Reviewer #1: No

Reviewer #3: **Yes: **Sofia Douzgou Houge

---

## [Author Response · Author response to Decision Letter 1]

21 Nov 2022

• The STR loci were generated by GangSTR, however, the data validation is not performed as required from genomic sequencing data. At least the STR data should be confirmed by a different approach/tool for analyzing STR lengths and the selected samples should be sequenced by Sanger sequencing for the further validation.

o Thank you for your response. All positions referred to have been now validated via Long Reads with subsets of those positions validated via Sanger Sequencing or gel electrophoresis. The wording in the Validation section and Supplemental Table S3 have been modified to reflect these additions.

• Thank you for asking me to review this manuscript. This indicates that expansion propensity varies depending on different haplotypes. I think that a study of ethnic diversity or of how a haplotype may correlate with a tendency to expansion is interesting. However we would require more detailed/systematic clinical data to see for example if these expansions may also explain some of the morbidity. But although the numbers of the cohorts mentioned in the manuscript are impressive, I am afraid that my expertise is not appropriate for making a judgment about the soundness of the Methods used.

o Thank you for your comments. While we agree that additional clinical data would be valuable in examining the morbidity of many of these patients, we wanted to focus on the trends we noticed when identifying Short Tandem Repeats. We intend to pursue this information in subsequent studies.

---

## [Decision Letter · Decision Letter 2]

7 Dec 2022

The Genomic landscape of short tandem repeats across multiple ancestries

PONE-D-22-22440R2

Dear Dr. Vijayaraghavan,

We’re pleased to inform you that your manuscript has been judged scientifically suitable for publication and will be formally accepted for publication once it meets all outstanding technical requirements.

Kind regards,

Alvaro Galli

Academic Editor

PLOS ONE

Additional Editor Comments (optional):

Reviewers' comments:

Reviewer's Responses to Questions

**Comments to the Author**

1. If the authors have adequately addressed your comments raised in a previous round of review and you feel that this manuscript is now acceptable for publication, you may indicate that here to bypass the “Comments to the Author” section, enter your conflict of interest statement in the “Confidential to Editor” section, and submit your "Accept" recommendation.

Reviewer #1: All comments have been addressed

Reviewer #3: All comments have been addressed

2. Is the manuscript technically sound, and do the data support the conclusions?

Reviewer #1: Yes

Reviewer #3: (No Response)

3. Has the statistical analysis been performed appropriately and rigorously? 

Reviewer #1: Yes

Reviewer #3: (No Response)

4. Have the authors made all data underlying the findings in their manuscript fully available?

Reviewer #1: Yes

Reviewer #3: (No Response)

5. Is the manuscript presented in an intelligible fashion and written in standard English?

Reviewer #1: Yes

Reviewer #3: (No Response)

6. Review Comments to the Author

Reviewer #1: This is a revised manuscript of "The Genomic landscape of short tandem repeats across multiple ancestries". I have no more comments.

Reviewer #3: (No Response)

7. PLOS authors have the option to publish the peer review history of their article (what does this mean?). If published, this will include your full peer review and any attached files.

Reviewer #1: No

Reviewer #3: No

---

## [Editor Report · Acceptance letter]

22 Dec 2022

PONE-D-22-22440R2 

The Genomic landscape of short tandem repeats across multiple ancestries     
Short tandem repeats and ancestry 

Dear Dr. Vijayaraghavan:

I'm pleased to inform you that your manuscript has been deemed suitable for publication in PLOS ONE. Congratulations! Your manuscript is now with our production department. 

Kind regards, 

on behalf of

Dr. Alvaro Galli 

Academic Editor

PLOS ONE